# Comparison of Oncologic Outcomes between Carbon Ion Radiotherapy and Stereotactic Body Radiotherapy for Early-Stage Non-Small Cell Lung Cancer

**DOI:** 10.3390/cancers13020176

**Published:** 2021-01-06

**Authors:** Yuhei Miyasaka, Shuichiro Komatsu, Takanori Abe, Nobuteru Kubo, Naoko Okano, Kei Shibuya, Katsuyuki Shirai, Hidemasa Kawamura, Jun-ichi Saitoh, Takeshi Ebara, Tatsuya Ohno

**Affiliations:** 1Department of Radiation Oncology, Gunma University Graduate School of Medicine, 3-39-15, Showa-Machi, Maebashi, Gunma 371-8511, Japan; m1720022@gunma-u.ac.jp (S.K.); kubo@gunma-u.ac.jp (N.K.); okano.n@gunma-u.ac.jp (N.O.); shibukei@gunma-u.ac.jp (K.S.); kawa@gunma-u.ac.jp (H.K.); tohno@gunma-u.ac.jp (T.O.); 2Gunma University Heavy Ion Medical Center, 3-39-15, Showa-Machi, Maebashi, Gunma 371-8511, Japan; 3Department of Radiation Oncology, Saitama Medical University International Medical Center, 1397-1, Yamane, Hidaka, Saitama 350-1241, Japan; t_abe@saitama-med.ac.jp; 4Department of Radiology, Jichi Medical University, 3311-1, Yakushiji, Shimotsuke, Tochigi 329-0498, Japan; kshirai@jichi.ac.jp; 5Department of Radiology, Division of Radiation Oncology, Faculty of Medicine, Academic Assembly, University of Toyama, 2630, Sugitani, Toyama, Toyama 930-0194, Japan; junsaito@med.u-toyama.ac.jp; 6Department of Radiation Oncology, Kyorin University, 6-20-2, Shinkawa, Mitaka, Tokyo 181-8611, Japan; tebara@ks.kyorin-u.ac.jp

**Keywords:** non-small cell lung cancer, radiotherapy, stereotactic body radiotherapy, carbon ion radiotherapy, outcomes, comparison

## Abstract

**Simple Summary:**

Lung cancer is a leading cause of cancer-related death. Stereotactic body radiotherapy (SBRT) is the standard treatment for inoperable early-stage non-small cell lung cancer (NSCLC). Carbon ion radiotherapy (CIRT) is a safe and effective treatment for early-stage NSCLC. However, there is no direct comparison study between these treatments. The present study aimed to compare oncologic outcomes after CIRT and SBRT for early-stage NSCLC in a single-institutional and contemporaneous cohort. We demonstrated favorable overall survival and local control in the CIRT group compared to those in the SBRT group using log-rank tests and Cox regression analyses for 89 patients. In addition, these results were validated in propensity score-adjusted analyses. The present retrospective comparison study showed a positive efficacy profile of CIRT, which is beneficial in the management of early-stage NSCLC.

**Abstract:**

Lung cancer is a leading cause of cancer-related deaths worldwide. Radiotherapy is an essential treatment modality for inoperable non-small cell lung cancer (NSCLC). Stereotactic body radiotherapy (SBRT) is the standard treatment for early-stage NSCLC because of its favorable local control (LC) compared to conventional radiotherapy. Carbon ion radiotherapy (CIRT) is a kind of external beam radiotherapy characterized by a steeper dose distribution and higher biological effectiveness. Several prospective studies have shown favorable outcomes. However, there is no direct comparison study between CIRT and SBRT to determine their benefits in the management of early-stage NSCLC. Thus, we conducted a retrospective, single-institutional, and contemporaneous comparison study, including propensity score-adjusted analyses, to clarify the differences in oncologic outcomes. The 3-year overall survival (OS) was 80.1% in CIRT and 71.6% in SBRT (*p* = 0.0077). The 3-year LC was 87.7% in the CIRT group and 79.1% in the SBRT group (*p* = 0.037). Multivariable analyses showed favorable OS and LC in the CIRT group (hazard risk [HR] = 0.41, *p* = 0.047; HR = 0.30, *p* = 0.040, respectively). Log-rank tests after propensity score matching and Cox regression analyses using propensity score confirmed these results. These data provided a positive efficacy profile of CIRT for early-stage NSCLC.

## 1. Introduction

Lung cancer is a leading cause of cancer-related deaths worldwide [1]. Lobectomy with mediastinal lymph node dissection is the standard treatment for early-stage non-small cell lung cancer (NSCLC), while radiotherapy plays an essential role in managing inoperable cases [2]. Stereotactic body radiotherapy (SBRT) contributes to better local control (LC) and overall survival (OS) compared with conventional radiotherapy [3]. Thus, SBRT has mostly replaced conventional radiotherapy as the standard approach for inoperable patients with early-stage NSCLC. Some studies showed that SBRT for operable stage I NSCLC was comparable with lobectomy [4]. Carbon ion radiotherapy (CIRT) is a type of external beam radiotherapy with a stronger tumor-cell-killing ability per physical dose due to its higher relative biological effect and sharper dose distribution than photon radiotherapy [5]. Previous studies demonstrated safe and effective outcomes of CIRT for early-stage NSCLC when compared with the literature on SBRT [6,7,8,9]. The safety of CIRT was also seen in older patients with interstitial lung disease [10,11,12]. These results were supported by a dosimetric analysis comparing CIRT and SBRT, showing that the dose distribution of CIRT exhibited better target conformity and spared organs at risk, including the lungs [13]. Comparison studies between CIRT and SBRT to confirm the benefits of CIRT are required. However, only literature comparisons are available, which makes the utility of CIRT vague. One reason for the lack of an institutional comparison study is that few facilities are available for performing both techniques. Thus, we analyzed the clinical outcomes of CIRT and SBRT for early-stage NSCLC in a single-institutional and contemporaneous cohort to compare the effectiveness of these treatments.

## 2. Results

### 2.1. Patient Characteristics of the Entire Cohort

A total of 89 patients meeting the inclusion criteria were identified in our database. Sixty-two patients underwent CIRT, and 27 underwent SBRT. Patient characteristics are summarized in Table 1. The numbers of younger patients and T2a diseases were significantly larger in the CIRT group. The median follow-up periods in the entire cohort, CIRT group, and SBRT group were 60.5 months (range, 2.0–120), 64.4 months (range, 2.0–120), and 39.1 months (range, 2.1–89.3), respectively.

### 2.2. Clinical Outcomes of the Entire Cohort

A total of 31 patients died during the final follow-up. The causes of death in the CIRT and SBRT groups were as follows: NSCLC, 15% (9/62) and 19% (5/27); noncancerous lung disease (e.g., pneumonitis, acute exacerbation of chronic obstructive pulmonary disease (COPD)) were 4.8% (3/62), and 15% (4/27); other causes or unknown were 9.7% (6/62) and 15% (4/27), respectively. Fisher’s exact tests were performed to test the difference between the CIRT and SBRT groups and showed no statistically significant difference (the NSCLC, *p* = 0.75; noncancerous lung disease, *p* = 0.19; the other or unknown, *p* = 0.48). Grade 3 or worse acute toxicity was not observed in the CIRT and SBRT groups. One patient in each treatment group (1.6% for CIRT and 3.7% for SBRT) developed grade 3 radiation pneumonitis. Grade 4 or worse late toxicity related with the treatment was not observed.

The 3-year and 5-year OS rates were 80.1% (95% confidence interval (95% CI), 70.7–90.9%) and 73.4% (95% CI, 63.0–85.5%) in the CIRT group, and 71.6% (95% CI, 54.7–90.3%) and 46.3% (95% CI, 28.8–74.3%) in the SBRT group, respectively. A log-rank test showed that CIRT was significantly associated with higher OS rates than SBRT (*p* = 0.0077) (Figure 1A). A Cox regression analysis also showed favorable OS in the CIRT group compared with the SBRT group (hazard risk (HR), 0.41 (95% CI, 0.17–0.99), *p* = 0.047) (Table 2).

The 3-year and 5-year progression-free survival (PFS) rates were 60.4% (95% CI, 49.3–74.1%) and 55.3% (95% CI, 44.0–69.4%) in the CIRT group, and 62.5% (95% CI, 44.9–87.2%) and 29.8% (95% CI, 14.8–60.0%) in the SBRT group (Figure 1B). These differences were not statistically significant (*p* = 0.14) (Figure 1B). A Cox regression analysis showed no statistically significant differences (Table 2).

The 3-year and 5-year LC rates were 87.7% (95% CI, 79.6–96.7%) and 85.5% (95% CI, 76.6–95.4%) in the CIRT group, and 79.1% (95% CI, 62.7–99.8%) and 55.3% (95% CI, 34.3–89.1%) in the SBRT group (Figure 1C). A log-rank test showed that CIRT was significantly associated with higher LC rates than SBRT (*p* = 0.037) (Figure 1C). A Cox regression analysis showed favorable LC in the CIRT group than in the SBRT group (HR, 0.30 (95% CI, 0.097–0.95), *p* = 0.040) (Table 2).

### 2.3. Propensity Score-Adjusted Analyses

Propensity scores were estimated by binomial logistic regression using available covariates, including age, sex, Eastern Cooperative Oncology Group Performance Status (ECOG PS), Charlson comorbidity index (CCI), smoking status, Brinkman index, the presence of COPD, presence of interstitial pneumonia (IP), T stage (UICC 7th edition), pathological diagnosis, and calendar year of treatment. The c-statistic value was 0.832. The distributions of propensity scores before and after matching are shown in Appendix A. Characteristics of the matched patients are summarized in Table 3. The median follow-up periods in the entire matched cohort, CIRT group, and SBRT group were 50.5 months (range, 5.9–117), 63.9 months (range, 14.6–117), and 26.8 months (range, 5.9–89.3), respectively.

Log-rank tests were performed in the matched cohort. The 3-year and 5-year OS rates were 93.3% (95% CI, 81.5–100%) and 71.8% (95% CI, 51.8–99.6%) in the CIRT group, and 55.0% (95% CI, 32.2–93.8%) and 34.4% (95% CI, 14.1–80.9%) in the SBRT group, respectively, which were statistically significantly different (*p* = 0.043) (Figure 2A).

The 3-year and 5-year PFS rates were 59.3% (95% CI, 38.7–90.7%) and 51.9% (95% CI, 31.4–85.5%) in the CIRT group, and 47.2% (95% CI, 25.3–88.2%) and 18.9% (95% CI, 5.45–65.4%) in the SBRT group, respectively, which were not statistically significantly different (*p* = 0.20) (Figure 2B).

The 3-year and 5-year LC rates were both 92.3% (95% CI, 78.9–100%) in the CIRT group, and 70.7% (95% CI, 47.2–100%) and 42.4% (95% CI, 18.7–96.5%) in the SBRT group, respectively, which were statistically significantly different (*p* = 0.022) (Figure 2C).

Finally, we performed Cox regression analyses using propensity scores as covariates. Favorable OS, PFS, and LC in the CIRT group were observed (HR, 0.34 (95% CI, 0.14–0.83%), *p* = 0.018; HR, 0.41 (95% CI, 0.19–0.88%), *p* = 0.022; HR, 0.19 (95% CI, 0.059–0.61%), *p* = 0.0051, respectively) (Table 4).

## 3. Discussion

To the best of our knowledge, this is the first report comparing the oncologic outcomes following CIRT and SBRT for early-stage NSCLC in a single-institutional and contemporaneous cohort. CIRT was statistically significantly associated with higher OS and LC rates than those of SBRT in log-rank tests. Cox regression analyses also showed favorable OS and LC in the CIRT group. In addition, propensity score-matching and Cox regression analyses using propensity scores as covariates certified these results. Grade 3 radiation pneumonitis was observed in one patient each in the two groups.

There are several prospective studies regarding SBRT for early-stage NSCLC. In the Japan Clinical Oncology Group (JCOG) 0403 study, 48 Gy in four fractions over 4 days were delivered to patients with T1N0M0 NSCLC, and the 3-year OS and PFS were 59.9% and 54.5%, respectively [14]. A recent meta-analysis estimated 3-year OS following SBRT to be 63% [15]. Our SBRT procedure was performed according to the JCOG 0403 study. Our results on SBRT showed 3-year OS and PFS rates of 71.6% and 62.5%, respectively, which are almost consistent with these studies.

CIRT for early-stage NSCLC was surveyed in a few prospective trials. In these studies, the 3-year OS and LC were 76–86% and 86–90%, respectively [6,7,8,9]. Our findings also seem to be consistent with these previous studies. Our single-institutional and contemporaneous comparison study validated the clinical advantages of CIRT over SBRT, previously estimated only by the literature comparisons.

It is noteworthy that the LC rate after CIRT was favorable despite disease progression. In our baseline cohort, the number of patients with T2a disease was significantly larger in the CIRT group than in the SBRT group (29% vs. 3.7%, *p* = 0.009). However, the LC rate after CIRT was significantly higher than in the SBRT group (5-year LC, 85.5% vs. 55.3%, *p* = 0.037). Cox regression and propensity score-adjusted analyses were consistent with this result. Previous studies supported our findings. In CIRT, the 2-year LC of T2b–4N0M0 NSCLC was 81% [16], which suggested that CIRT was effective even for larger tumors. In contrast, local failures after SBRT were observed in 25% (7/28) of the patients with T2N0M0 NSCLC [17], while 12% (20/169) of T1N0M0 disease developed local failure [14]. These results suggest that LC following SBRT could decrease in patients with larger tumors. Taken together, the superiority of CIRT in LC may be more apparent when treating larger tumors.

Furthermore, OS was favorable in CIRT in the present study. However, a previous systematic review and meta-analysis for Stage I NSCLC showed no statistically significant difference in OS between CIRT and SBRT [18]. A possible explanation for these results could be that the literature on CIRT included in their systematic review had a limited population from a single institution. Additionally, only medical inoperability was adjusted as a confounder in their study, but other factors, such as age, performance status, histologic type, and tumor size, which were considered prognostic factors in other studies [8,19,20], were not adjusted in their research. This situation may have affected their results. In our study, the numbers of younger patients and locally advanced tumors were significantly larger in the CIRT group. To eliminate potential bias derived from prognostic factors, we performed propensity score-adjusted analyses, which showed the survival advantages of CIRT.

When interpreting our findings, we should consider the difference in the prescribed clinical doses between CIRT and SBRT. In the present study, CIRT of 52.8 or 60.0 Gy (relative biological effectiveness (RBE)) and SBRT of 48 Gy were delivered. At first glance, the higher dose of CIRT seemed to be one of the reasons for the favorable outcomes following CIRT. However, we should acknowledge that the Gy (RBE) does not always represent a biologically equivalent dose to the absorbed dose in photon radiotherapy. In clinical practice using CIRT, a fixed RBE of 3.0 is generally utilized for convenience. However, the actual RBE could be affected by various factors including linear energy transfer, tissue types, and endpoints of concern, which makes it difficult to determine the exact equivalent dose of CIRT to photon radiotherapy. Accordingly, we should not conclude that the favorable outcomes were caused by the difference in the prescribed clinical doses. In future studies, including a randomized controlled trial comparing CIRT and SBRT, the prescribed doses should be considered based on the biological and physical aspects.

There are some limitations to our study. First, we cannot exclude potential sources of bias due to unmeasured confounders, even with propensity score-adjusted analyses (e.g., medical operability, respiratory functions, gene mutations, distance to hospital, income, and other unknown confounders). Second, the lack of information regarding medical operability status made it difficult to compare surgical treatment and CIRT/SBRT. We consider that these comparisons are future issues, since these are required in the management of operable patients. Finally, dose escalation may improve the prognosis of SBRT, as shown in a previous meta-analysis [21]. In addition, there are various SBRT procedures regarding prescribed dose fractionations, prescribed points, gating, and treatment devices. Optimizing these factors may lead to a better prognosis.

## 4. Materials and Methods

### 4.1. Study Design

The present study is a retrospective cohort study that compared oncologic outcomes after CIRT and SBRT for early-stage peripheral NSCLC. We identified patients who met the inclusion criteria from our database and evaluated OS, PFS, and LC rates after these treatments. Univariate and multivariable analyses were performed to test differences in these outcomes. In order to make these results more robust, we performed propensity score-adjusted analyses, including propensity score-matching and multivariable analyses using propensity scores as covariates.

### 4.2. Patients

From our database, we identified patients who met the following criteria: (i) pathologically or clinically diagnosed NSCLC; (ii) clinical T1a–2aN0M0 (Union for International Cancer Control (UICC), 7th edition) and peripheral disease; (iii) treated with CIRT or SBRT at Gunma University Hospital (Gunma University Heavy Ion Medical Center) between 2010 and 2015. Each patient chose between CIRT and SBRT, based on the patient’s preference after being informed. The appropriateness of the treatment was confirmed by the institutional cancer board, attended by pulmonologists, respiratory surgeons, radiation oncologists, and diagnostic radiologists.

### 4.3. Carbon Ion Radiotherapy

Details of the CIRT technique have been reported previously [9]. Briefly, the patient is immobilized in the supine or prone position with rotations of ±15° to the superior–inferior axis with a thermoplastic shell (Shellfitter; Sanyo Polymer Industrial, Nara, Japan) and a pillow made of water-sclerogenic polymers (Moldcare; ALCARE, Tokyo, Japan). A respiratory-gated computerized tomography (CT) image with 2 mm thick slices was taken after exhaling. A four-dimensional CT scan was also performed to assess respiratory motion and reconstruct four-dimensional images of each respiration phase. The gross tumor volume (GTV) was delineated in the lung window. The clinical target volume (CTV) was generated by adding a 5 mm margin into the pulmonary parenchyma to the GTV. The inner margin was determined from the four-dimensional CT, with a 3 mm setup margin. A margin calculated as the square root of the sum of squares of the inner and setup margins was added to the CTV to create the planning target volume (PTV). The clinical dose distribution was based on the physical dose multiplied by the relative biological effectiveness (RBE) of 3.0, according to a previous report [5]. XiO-N treatment-planning software (Elekta/Mitsubishi Electric) was used to calculate the passive scattering carbon-ion dose distribution. The total prescribed dose was 52.8 Gy or 60.0 Gy (RBE) for the isocenter in four fractions over a week, with a fractional dose of 13.2 Gy or 15.0 Gy (RBE) at four treatment sessions per week. Respiratory-gated irradiation with a gating level of <30% of the wave height around peak exhalation was applied at each treatment session.

### 4.4. Stereotactic Body Radiotherapy

SBRT was performed according to the protocol of the JCOG 0403 study [14]. CT images with 2.5 mm thick slices were taken during normal quiet breathing, and four-dimensional CT images were also taken. The inner margin was estimated as the sum of the primary tumor in each phase. The CTV was defined as the inner margin plus a 5 mm margin in all lung parenchyma directions. The PTV was defined as CTV plus a 3 mm margin for setup. The isocenter was located at the PTV center, and the dose was prescribed to the isocenter. The total dose was 48 Gy in four fractions a week, with a fractional dose of 12.0 Gy.

### 4.5. Follow-Up and Evaluation

After the treatment, the patients were followed up with blood tests, including tumor markers and CT scans or chest X-rays. In the case of patients who refused regular follow-ups at our hospital, their clinical course was obtained via telephone, letters, and patient referral documents. According to the Response Evaluation Criteria in Solid Tumors guidelines version 1.1 [22], the tumor response to the treatment was evaluated. Adverse events were evaluated based on the Common Terminology Criteria for Adverse Events (CTCAE) version 5.0 [23]. OS, PFS, and LC were defined as the periods from the first day of irradiation to death from any cause, disease progression at any site or death from any cause, and tumor regrowth or recurrence in the PTV, respectively, or the last follow-up. When considering LC, patients who died were censored at the date of death.

### 4.6. Statistical Analysis

We calculated OS, PFS, and LC rates from the first day of irradiation using the Kaplan–Meier method. The log-rank test was used to evaluate the associations of these endpoints with the treatment modalities. These endpoints were compared between subgroups using the Cox proportional hazards regression model to estimate the HR and its 95% CI. Covariates were determined based on prognostic factors proposed in the literature and the number of events.

Propensity score-adjusted analyses were performed to minimize overt selection bias. Propensity scores were estimated by binomial logistic regression using available covariates, including age, sex, ECOG PS, CCI, smoking status, Brinkman index, the presence of COPD, the presence of IP, T stage, pathological diagnosis, and calendar year of treatment. Propensity score matching was performed by one-to-one nearest-neighbor matching using a caliper width of 0.25 times the pooled standard deviation of the propensity scores’ logit without replacement. Cox regression analyses using propensity scores as covariates were also performed to make our results more robust.

A *p*-value < 0.05 was considered statistically significant. All statistical analyses and visualization were performed using R 3.6.2 [24].

## 5. Conclusions

In summary, a favorable efficacy profile of CIRT for early-stage NSCLC compared with SBRT was shown via this single-institutional and contemporaneous comparison study. For more reliable evidence, a randomized controlled trial is warranted in future.

## Figures and Tables

**Figure 1 cancers-13-00176-f001:**
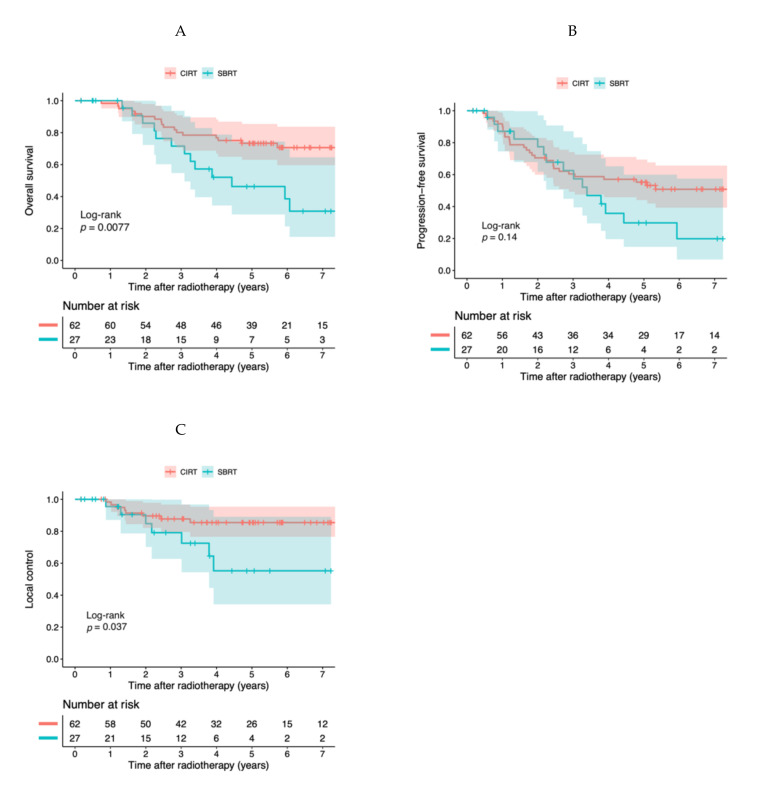
Kaplan–Meier curves for the entire cohort (*n* = 89). (**A**) Overall survival, (**B**) progression-free survival, and (**C**) local control. Colored regions show 95% confidence intervals.

**Figure 2 cancers-13-00176-f002:**
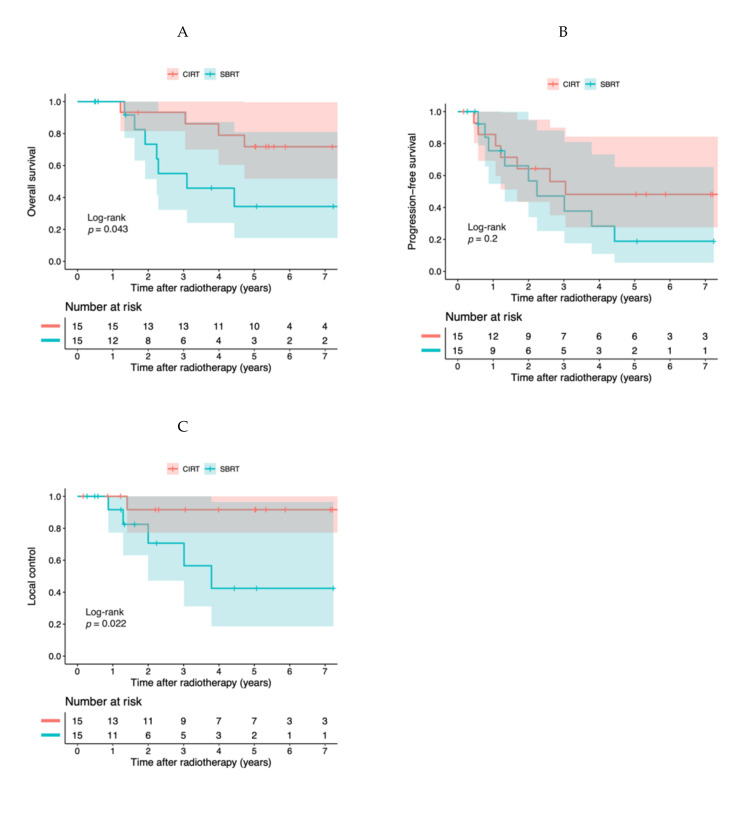
Kaplan–Meier curved for the matched cohort (*n* = 30). (**A**) Overall survival, (**B**) progression-free survival, and (**C**) local control. Colored regions show 95% confidence intervals.

**Table 1 cancers-13-00176-t001:** Patient characteristics of the entire cohort (*n* = 89).

Characteristic	Overall, *n* = 89 ^1^	CIRT, *n* = 62 ^1^	SBRT, *n* = 27 ^1^	*p*-Value ^2^
Age	74 (66,80)	72 (64,79)	77 (73,82)	0.032
Sex				0.8
Male	63 (71%)	43 (69%)	20 (74%)	
Female	26 (29%)	19 (31%)	7 (26%)	
ECOG PS				>0.9
0	42 (47%)	29 (47%)	13 (48%)	
1	45 (51%)	31 (50%)	14 (52%)	
2	2 (2.2%)	2 (3.2%)	0 (0%)	
CCI				0.12
0	36 (40%)	28 (45%)	8 (30%)	
1	11 (12%)	8 (13%)	3 (11%)	
2	23 (26%)	16 (26%)	7 (26%)	
3	10 (11%)	5 (8.1%)	5 (19%)	
4	6 (6.7%)	5 (8.1%)	1 (3.7%)	
5	1 (1.1%)	0 (0%)	1 (3.7%)	
6	2 (2.2%)	0 (0%)	2 (7.4%)	
Pathological diagnosis				>0.9
Adenocarcinoma	48 (54%)	33 (53%)	15 (56%)	
Squamous cell carcinoma	26 (29%)	19 (31%)	7 (26%)	
Clinical	15 (17%)	10 (16%)	5 (19%)	
T stage				0.009
1a	41 (46%)	23 (37%)	18 (67%)	
1b	29 (33%)	21 (34%)	8 (30%)	
2a	19 (21%)	18 (29%)	1 (3.7%)	
Brinkman Index	572 (0,1,120)	511 (0,1,085)	600 (50,1,335)	0.2
Smoking				0.3
Never	32 (36%)	25 (40%)	7 (26%)	
Past/ Current	57 (64%)	37 (60%)	20 (74%)	
COPD				0.7
No	62 (70%)	42 (68%)	20 (73%)	
Yes	27 (30%)	20 (32%)	7 (26%)	
Interstitial pneumonia				0.2
No	86 (97%)	61 (98%)	25 (93%)	
Yes	3 (3.4%)	1 (1.6%)	2 (7.4%)	
Treatment year				0.065
2010–2012	42 (47%)	25 (9.7%)	17 (3.7%)	
2013–2015	47 (53%)	37 (9.7%)	10 (3.7%)	

^1^ Statistics presented: median (IQR); *n* (%); ^2^ Statistical tests performed: Wilcoxon rank-sum test; chi-square test of independence; Fisher’s exact test; Abbreviations: CIRT, carbon ion radiotherapy; SBRT, stereotactic body radiotherapy; ECOG PS, Eastern Cooperative Oncology Group Performance Status; CCI, Charlson Comorbidity Index; COPD, chronic obstructive pulmonary disease.

**Table 2 cancers-13-00176-t002:** Cox regression analysis (*n* = 89).

Variables	Subgroup	Overall Survival		Progression-Free Survival		Local Control	
		HR (95% CI)	*p*-Value	HR (95% CI)	*p*-Value	HR (95% CI)	*p*-Value
Treatment	SBRT	reference	0.047	reference	0.20	reference	0.040
	CIRT	0.41 (0.17–0.99)		0.61 (0.29–1.3))		0.30 (0.097–0.95)	
T stage	T1a–1b	reference	0.00095	reference	0.0023	reference	0.48
	T2a	5.5 (2.0–15)		3.2 (1.5–6.8)		1.7 (0.4–6.9)	
CCI	0–2	reference	0.00015	reference	0.019		
	3–6	6.0 (2.4–15.3)		2.5 (1.2–5.5)			

CIRT, carbon ion radiotherapy; SBRT, stereotactic body radiotherapy; CCI, Charlson Comorbidity Index.

**Table 3 cancers-13-00176-t003:** Patient characteristics of the matched cohort (*n* = 30).

Characteristic	Overall, *n* = 30 ^1^	CIRT, *n* = 15 ^1^	SBRT, *n* = 15 ^1^	*p*-Value ^2^
Age	76 (70,82)	74 (66,82)	76 (71,79)	>0.9
Sex				>0.9
Male	23 (77%)	11 (73%)	12 (80%)	
Female	7 (23%)	4 (27%)	3 (20%)	
ECOG PS				0.7
0	14 (47%)	8 (53%)	6 (40%)	
1	16 (53%)	7 (47%)	9 (60%)	
2	0 (0%)	0 (0%)	0 (0%)	
CCI				0.6
0	8 (27%)	4 (27%)	4 (27%)	
1	4 (13%)	3 (20%)	1 (6.7%)	
2	8 (27%)	3 (20%)	5 (33%)	
3	7 (23%)	3 (20%)	4 (27%)	
4	2 (6.7%)	2 (13%)	0 (0%)	
5	0 (0%)	0 (0%)	0 (0%)	
6	1 (3.3%)	0 (0%)	1 (6.7%)	
Pathological diagnosis				>0.9
Adenocarcinoma	15 (50%)	8 (53%)	7 (47%)	
Squamous cell carcinoma	11 (37%)	5 (33%)	6 (40%)	
Clinical	4 (13%)	2 (13%)	2 (13%)	
T stage				>0.9
1a	15 (50%)	7 (47%)	8 (53%)	
1b	13 (43%)	7 (47%)	6 (40%)	
2a	2 (6.7%)	1 (6.7%)	1 (6.7%)	
Brinkman Index	778 (0, 1,100)	760 (33, 950)	900 (0, 1,415)	0.6
Smoking				>0.9
Never	9 (30%)	4 (27%)	5 (33%)	
Past/Current	21 (70%)	11 (73%)	10 (67%)	
COPD				>0.9
No	22 (73%)	11 (73%)	11 (73%)	
Yes	8 (27%)	4 (27%)	4 (27%)	
Interstitial pneumonia				>0.9
No	30 (100%)	15 (100%)	15 (100%)	
Yes	0 (0%)	0 (0%)	0 (0%)	
Treatment year				0.26
2010–2012	12 (40%)	4 (33%)	8 (67%)	
2013–2015	18 (60%)	11 (61%)	7 (39%)	

^1^ Statistics presented: median (IQR); *n* (%); ^2^ Statistical tests performed: Wilcoxon rank-sum test; Fisher’s exact test; chi-square test of independence; Abbreviations: CIRT, carbon ion radiotherapy; SBRT, stereotactic body radiotherapy; ECOG PS, Eastern Cooperative Oncology Group Performance Status; CCI, Charlson Comorbidity Index; COPD, chronic obstructive pulmonary disease.

**Table 4 cancers-13-00176-t004:** Cox regression analysis using the propensity scores as covariates (*n* = 89).

Variables	Subgroup	Overall Survival		Progression-Free Survival		Local Control	
		HR (95% CI)	*p*-Value	HR (95% CI)	*p*-Value	HR (95% CI)	*p*-Value
Treatment	SBRT	reference	0.018	reference	0.022	reference	0.0051
	CIRT	0.34 (0.14–0.83)		0.41 (0.19–0.88)		0.19 (0.059–0.61)	
Propensity scores		1.4 (0.31–6.4)	0.66	3.3 (0.86–13)	0.082	8.0 (0.79–82)	0.78

CIRT, carbon ion radiotherapy; SBRT, stereotactic body radiotherapy.

## Data Availability

The supporting data are not publicly available due to their containing information that could compromise the privacy of research participants.

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
