# Peer review of "Comparison of Oncologic Outcomes between Carbon Ion Radiotherapy and Stereotactic Body Radiotherapy for Early-Stage Non-Small Cell Lung Cancer"

_cancers, 2021, doi:10.3390/cancers13020176_

Round 1
Reviewer 1 Report
The article is original and adresses a well defined problem. To the best of my knowledge, there so far have been made no direct comparative studies that would analyze CIRT and SBRT.
The results in terms of OS and LC look promising. I expect more studies of such kind that would entail larger numbers of treated patients.
By the way, I have noticed a discrepancy in data on LC rate 3-year in CIRT group: line 44 indicates that the rate is 85.5%, but lines 96 and 97 suggest that 3-year LC in CIRT group is 87.7% and 5-year LC is 85.5%. I assume this should be corrected.
It is widely recognized that CIRT is safer for patients with adverse conditions, such as large tumor. I suppose, the author shares this view by choosing a bigger number of T2a cases for the CIRT group (line 73). The usual assumption is that CIRT is also safer for the central tumor. However, the article does not contain sufficient information with regard to the tumor location. Overall, the results are interpreted appropriately and statistical analysis made carefully.
The article is written in an appropriate way. All the data is presented in a comprehensive manner. What captures me most is that the study is original. Patient characteristics, radiation delivery technique and statistical analysis are described with sufficient details to allow another researcher to reproduce the results.
Furthermore, the section “Discussion” takes into account all strengths and weaknesses of this study. I definitely agree with the author's view that the dose escalation in SBRT patients may improve the LC. For instance, the RTOG 0618 study have revealed a result of 97.6%.
Author Response
The article is original and adresses a well defined problem. To the best of my knowledge, there so far have been made no direct comparative studies that would analyze CIRT and SBRT.
The results in terms of OS and LC look promising. I expect more studies of such kind that would entail larger numbers of treated patients.
Point 1: By the way, I have noticed a discrepancy in data on LC rate 3-year in CIRT group: line 44 indicates that the rate is 85.5%, but lines 96 and 97 suggest that 3-year LC in CIRT group is 87.7% and 5-year LC is 85.5%. I assume this should be corrected.
Response 1: We appreciate your comment. In accordance with your comment, we have corrected the error.
Correction 1: (Line 44)
The 3-year LC was 87.7% in the CIRT group
Point 2: It is widely recognized that CIRT is safer for patients with adverse conditions, such as large tumor. I suppose, the author shares this view by choosing a bigger number of T2a cases for the CIRT group (line 73). The usual assumption is that CIRT is also safer for the central tumor. However, the article does not contain sufficient information with regard to the tumor location. Overall, the results are interpreted appropriately and statistical analysis made carefully.
Response 2: Thank you for the valuable comment. In the present study, only peripheral tumors were eligible. We added this information to the Materials and Methods section.
Correction 2: (Line 226–227)
(ii) clinical T1a–2aN0M0 (Union for International Cancer Control (UICC), 7th edition) and peripheral disease
Point 3: The article is written in an appropriate way. All the data is presented in a comprehensive manner. What captures me most is that the study is original. Patient characteristics, radiation delivery technique and statistical analysis are described with sufficient details to allow another researcher to reproduce the results.
Furthermore, the section “Discussion” takes into account all strengths and weaknesses of this study. I definitely agree with the author's view that the dose escalation in SBRT patients may improve the LC. For instance, the RTOG 0618 study have revealed a result of 97.6%.
Response 3: Thank you for your comment.
Reviewer 2 Report
This is a well written paper dealing with a most interesting topics
I am in favor of publication with only one minor change
In the discussion the authors should discuss in greater detail the issue of total dose.
the authors are claiming that CIRT results are better than SBRT, this is shown in the retrospective population and is confirmed by the matched pair analysis. The authors suggest to test this hypothesis in a prospective study.
The dose prescribed to the CIRT group was higher (52.8-60 Gy RBE) than the dose prescribed to the SBRT group (48 Gy). The careful reader will notice that the difference in outcome may be due to this difference in dose rather than to the difference in radiation modality. Even the careful reader, unless he has a specific knowledge of CIRT, may fail to appreciate that the RBE model employed by the authors was initially designed for dose per fraction around 2.7 Gy RBE and that it is used pragmatically disregarding the dependency of RBE on dose per fraction. As a result of this the 60 Gy RBE of CIRT in 4 fraction of 15 Gy RBE are not equivalent to 60 Gy of SBRT for any clinically measurable endpoint. As consequence of this comparing CIRT and SBRT is difficult. The authors should explain the issue and discuss how does it affect the interpretation of their results and the design of future trials and they should briefly report the toxicity of these 89 patients. Only a comparable or better toxicity profile in the CIRT arm can validate the claim that the observed advantage has to do with the modality and not with the
Author Response
Point 1: This is a well written paper dealing with a most interesting topics
I am in favor of publication with only one minor change
In the discussion the authors should discuss in greater detail the issue of total dose.
the authors are claiming that CIRT results are better than SBRT, this is shown in the retrospective population and is confirmed by the matched pair analysis. The authors suggest to test this hypothesis in a prospective study.
The dose prescribed to the CIRT group was higher (52.8-60 Gy RBE) than the dose prescribed to the SBRT group (48 Gy). The careful reader will notice that the difference in outcome may be due to this difference in dose rather than to the difference in radiation modality. Even the careful reader, unless he has a specific knowledge of CIRT, may fail to appreciate that the RBE model employed by the authors was initially designed for dose per fraction around 2.7 Gy RBE and that it is used pragmatically disregarding the dependency of RBE on dose per fraction. As a result of this the 60 Gy RBE of CIRT in 4 fraction of 15 Gy RBE are not equivalent to 60 Gy of SBRT for any clinically measurable endpoint. As consequence of this comparing CIRT and SBRT is difficult. The authors should explain the issue and discuss how does it affect the interpretation of their results and the design of future trials and they should briefly report the toxicity of these 89 patients. Only a comparable or better toxicity profile in the CIRT arm can validate the claim that the observed advantage has to do with the modality and not with the
Response 1: We sincerely appreciate your valuable comment. In accordance with your suggestions, we have revised the presentation and discussion of the prescribed dose and toxicities in our manuscript as follows:
Correction 1-1: (Line 192–196)
When interpreting our findings, we should consider the difference in the prescribed clinical doses between CIRT and SBRT. In the present study, a CIRT of 52.8 or 60.0 Gy (RBE) and SBRT of 48 Gy were delivered. At first glance, the higher dose of CIRT seemed to be one of the reasons for the favorable outcomes following CIRT. However, we should acknowledge that the Gy (RBE) does not always represent a biologically equivalent dose to the absorbed dose in photon radiotherapy. In clinical practice using CIRT, a fixed RBE of 3.0 is generally utilized for convenience. However, the actual RBE could be affected by various factors including linear energy transfer, tissue types, and endpoints of concern, which makes it difficult to determine the exact equivalent dose of CIRT to photon radiotherapy. Accordingly, we should not conclude that the favorable outcomes were caused by the difference in the prescribed clinical doses. In future studies including a randomized controlled trial comparing CIRT and SBRT, the prescribed doses should be considered based on the biological and physical aspects.
Correction 1-2:
(Line 91–93)
Grade 3 or worse acute toxicity was not observed. One patient each in the CIRT and SBRT groups (1.6% and 3.7%, respectively) developed grade 3 radiation pneumonitis. Grade 4 or worse late toxicity related to the treatment was not observed.
(Line 159–160)
Grade 3 radiation pneumonitis was observed in one patient each in the two groups
(Line 270–271)
Adverse events were evaluated based on the Common Terminology Criteria for Adverse Events (CTCAE) version 5.0 [23].
Reviewer 3 Report
Miyasaka and colleagues have done a great job in comparing a new radiotherapy strategy to standard of care in early stage lung cancer in inoperable patients.
However I do have some remarks:
In the introduction it should be more stressed that SBRT is meant for stage I-IIA NSCLC in inoperable patients and that it should be a discussion whether to treat with SBRT or surgery. Because this is the group that is studies in this paper.- In the part of the methods, the nature of the study is not clear. It seems as this is a retrospective case-control study. However it is strange that many more patients have had the CIRT instead of SBRT. How were patients informed and how did they chose between therapies is not clear to me.
- The fact that the methodology is not clear, I am not able to give optimal comments about the result part. It seems as if the results are clear and that comparing the groups is equally performed in the matched cohort. However it is not clear where the matched cohort consists of.
- In the discussion part, I miss items about comparison of surgery and SBRT and how this is shown in CIRT.
Author Response
Miyasaka and colleagues have done a great job in comparing a new radiotherapy strategy to standard of care in early stage lung cancer in inoperable patients.
However I do have some remarks:
Point 1: In the introduction it should be more stressed that SBRT is meant for stage I-IIA NSCLC in inoperable patients and that it should be a discussion whether to treat with SBRT or surgery. Because this is the group that is studies in this paper.
Response 1: Thank you for valuable comment. We added descriptions of the SBRT and surgery to the revised manuscript.
Correction 1: (Line 58–60)
Thus, SBRT has mostly replaced conventional radiotherapy as the standard approach for inoperable patients with early-stage NSCLC. Some studies showed that SBRT for operable stage I NSCLC was comparable with lobectomy [4]
Point 2: In the part of the methods, the nature of the study is not clear. It seems as this is a retrospective case-control study. However it is strange that many more patients have had the CIRT instead of SBRT. How were patients informed and how did they chose between therapies is not clear to me.
Response 2: The present study is a retrospective cohort study. Thus, the difference in the number of patients between the subgroups was not designed. All the patients were informed of both treatments and chose one based on the patient’s preference. The appropriateness of the treatment was confirmed by the institutional cancer board attended by pulmonologists, respiratory surgeons, radiation oncologists, and radiologists. A possible explanation of the larger number of patients in CIRT is as follows: since CIRT is a limited medical resource (there are only 6 CIRT facilities in Japan), but SBRT is a commonly available treatment modality, the number of patients referred to our facilities became large in CIRT. We added this information to the Materials and Methods section.
Correction 2: (Line 228–231)
Each patient chose between CIRT and SBRT, based on the patient’s preference after being informed. The appropriateness of the treatment was confirmed by the institutional cancer board that was attended by pulmonologists, respiratory surgeons, radiation oncologists, and diagnostic radiologists.
Point 3: The fact that the methodology is not clear, I am not able to give optimal comments about the result part. It seems as if the results are clear and that comparing the groups is equally performed in the matched cohort. However it is not clear where the matched cohort consists of.
Response 3: Thank you very much. We would like to explain the methodology of the present study. First, we performed univariate and multivariable analyses for the entire cohort (n = 89). Second, in order to make these results more robust, we performed two propensity score-adjusted analyses. One analysis used univariate analyses in the propensity score-matched cohort (n = 30). The other analyses used multivariable analyses using propensity scores as covariates for the entire cohort (n = 89). These propensity score-adjusted analyses complement the results of the first analyses. We have added a new section ‘Study design’ in the Materials and Methods with this information.
Correction 3: (Line 216–223)
The present study is a retrospective cohort study that compared oncologic outcomes after CIRT and SBRT for early-stage peripheral NSCLC. We identified patients who met inclusion criteria from our database and evaluated OS, PFS, and LC rates after these treatments. Univariate and multivariable analyses were performed to test differences in these outcomes. In order to make these results more robust, we performed propensity score-adjusted analyses including propensity score-matching and multivariable analyses using propensity scores as covariates.
Point 4: In the discussion part, I miss items about comparison of surgery and SBRT and how this is shown in CIRT.
Response 4: Thank you for your comment. We agree with your opinion that we should discuss comparisons of surgery and SBRT/CIRT. Furthermore, we understand these comparisons are needed in the clinical setting. However, we feel that a discussion regarding this issue is out of the scope in the present study, since our cohort included both inoperable and operable (i.e., refused surgical treatment) patients, which made it difficult to compare outcomes of CIRT/SBRT and surgery. We have revised the Discussion to reflect these limitations of our study.
Correction 4: (Line 209–211)
Second, the lack of information regarding medical operability status made it difficult to compare surgical treatment and CIRT/SBRT. We consider that these comparisons are future issues, since these are required in the management of operable patients.
Round 2
Reviewer 3 Report
In my opinion the nature of the study is more clear and can be interpreted better.
In this perspective the results are clear and support the conclusions drawn